# Antibiotic Resistance Patterns of Uropathogens Causing Urinary Tract Infections in Children with Congenital Anomalies of Kidney and Urinary Tract

**DOI:** 10.3390/children8070585

**Published:** 2021-07-08

**Authors:** Raluca Isac, Diana-Georgiana Basaca, Ioana-Cristina Olariu, Ramona F. Stroescu, Andrada-Mara Ardelean, Ruxandra M. Steflea, Mihai Gafencu, Adela Chirita-Emandi, Iulia Cristina Bagiu, Florin George Horhat, Dan-Dumitru Vulcanescu, Dan Ionescu, Gabriela Doros

**Affiliations:** 1IIIrd Pediatric Clinic, Department of Pediatrics, “Victor Babeș” University of Medicine and Pharmacy Timișoara, 300041 Timișoara, Romania; isac.raluca@umft.ro (R.I.); olariu.cristina@umft.ro (I.-C.O.); andradamara@gmail.com (A.-M.A.); steflea.ruxandra@umft.ro (R.M.S.); mgafencu@umft.ro (M.G.); gdoros@gmail.com (G.D.); 2Emergency Hospital for Children “Louis Turcanu”, 300011 Timișoara, Romania; diana.basaca@umft.ro (D.-G.B.); stroescu.ramona@umft.ro (R.F.S.); adela.chirita@umft.ro (A.C.-E.); dannvulcanescu@gmail.com (D.-D.V.); 3Ist Pediatric Clinic, Department of Pediatrics, “Victor Babeș” University of Medicine and Pharmacy Timișoara, 300041 Timișoara, Romania; 4Department of Microscopic Morphology, Genetics Discipline, Center of Genomic Medicine, “Victor Babes” University of Medicine and Pharmacy, 300041 Timișoara, Romania; 5Multidisciplinary Research Center on Antimicrobial Resistance (MULTI-REZ), Microbiology Department, Victor Babes University of Medicine and Pharmacy, 300041 Timișoara, Romania; 6Physical Education and Sports Department, Polytechnic University, 300223 Timișoara, Romania; dan.ionescu@upt.ro

**Keywords:** antibiotic resistance, urine culture, drug-resistant bacteria, children, abnormal urinary tract

## Abstract

**Background:** Urinary tract infections (UTI) are common in children worldwide. Congenital anomalies of kidney and urinary tract (CAKUT) increase the risk of UTI and consequently antibiotic resistance. Antibiotic resistance represents an important public health issue worldwide. We aimed to evaluate the local trend in terms of bacterial uropathogen resistance in the western part of Romania in children with CAKUT and UTI. **Methods:** 252 children with CAKUT were admitted to our hospital over a five-year period. Of them, 91 developed at least one UTI episode, with a total number of 260 positive urine cultures. We collected data about age at diagnosis of CAKUT, sex, origin environment, type and side of CAKUT, number of UTIs, type of uropathogen, and uropathogens antibiotic resistance. **Results:** Distribution of uropathogens was *Escherichia coli* (38.84%), *Klebsiella* spp. (21.15%), *Enterococcus* spp. (15.76%), *Proteus* spp. (8.07%), *Pseudomonas* spp. (8.07%), *Enterobacter* spp. (2.3%), other Gram-negative bacteria (2.3%), and other Gram-positive bacteria (3.45%). High antibiotic resistance was detected for ampicillin, amoxicillin, and second-generation cephalosporins. *Escherichia coli* presented high resistance for cefepime and ceftriaxone. *Pseudomonas* spp. remained susceptible to amikacin, quinolones, and colistin. Vancomycin, teicoplanin, linezolid, and piperacillin/tazobactam remained effective in treating Gram-positive UTI. **Conclusions:** High antibiotic resistance was identified for frequently used antibiotics. Lower antibiotic resistance was observed for some broad-spectrum antibiotics. Understanding uropathogens’ antibiotic resistance is important in creating treatment recommendations, based on international guidelines, local resistance patterns, and patient particularities.

## 1. Introduction

Urinary tract infections are one of the most common infections in children, with an estimated occurrence between 10–30% of cases [1]. Some children are at increased risk of developing urinary tract infection (UTI) due to predisposing conditions including congenital abnormalities of the urinary tract favoring urine stasis [2]. In 30% of children with congenital anomalies of kidney and urinary tract (CAKUT), UTI can be the sentinel event [3]. CAKUT are a well-documented risk factor for the development of UTI in children [3]. Vesicoureteral reflux (VUR) alters the unidirectional flow of urine, while pyelo-ureteral junction obstruction (PUJO) leads to stasis, both increasing the risk of multiplying pathogenic microorganisms [4].

Any pathogen microorganism can cause UTI, yet bacteria are responsible for more than 90% of the cases [5]. *Escherichia coli* (*E. coli*) is the most common organism causing UTI in children [6]. Apart from that, the Infectious Diseases Society of America has high-lightened a section of antibiotic-resistant bacteria (*Enterococcus faecium, Staphylococcus aureus, Klebsiella pneumoniae, Acinetobacter baumannii, Pseudomonas aeruginosa* and *Enterobacter* spp.) under the acronym “the ESKAPE pathogens”, capable of “escaping” the bactericidal action of antibiotics [7,8]. This group of bacteria represents a challenge for empirical antibiotic treatment of complicated UTI [7,8,9].

Treatment of UTI begins immediately after a urine specimen for culture is collected and before the culture results are available and then changing to specific therapy based on uropathogens sensibility [9,10,11,12]. The treatment aims to eradicate the infection, prevent bacteriemia, relieve the symptoms, improve the clinical condition, avert complications, and prevent renal scarring [12,13]. Treatment should include 7–14 days of antimicrobial usage [13]. Guidelines for empirical antibiotic treatment of complicated UTI recommend amoxicillin/clavulanic acid or cefotaxime/ceftriaxone administered intravenously and then switching to oral as soon as the clinical condition permits it. The treatment duration must be 10 days for pyelonephritis and 14 days for urosepsis [9,12,14]. Bacterial resistance is a public health issue all over the world. Specifically, local antibiotic resistance patterns, if available, are useful in the selection of the initial antibiotic [12,14].

The aim of the study was to evaluate regional resistance patterns of uropathogens causing UTI in children with CAKUT, in the west part of Romania.

## 2. Materials and Methods

### 2.1. Study Design

The present retrospective study was conducted in the Emergency Hospital for Children "Louis Turcanu" in Timisoara, Romania between 2016 and 2020. The study design was approved by the Ethical Committee of the Emergency Hospital for Children "Louis Turcanu" under decision number 93/2020, and registry number 16913/11DEC2020.

We analyzed all patients diagnosed with CAKUT admitted to Emergency Hospital for Children "Louis Turcanu" in Timisoara over a 5-year period, aged <18 years (age of diagnosis) that had at least one episode of urinary tract infection. Urinary tract infection was defined by evidence from urinalysis and culture along with symptoms (fever, abdominal pain, chills, nausea, vomiting). Patients with CAKUT but with sterile urine culture were excluded from the study.

### 2.2. Participants

CAKUT was defined by abdominal ultrasonography in all patients and, if recommended, by computer tomography urography and/or voiding cystourethrogram. CAKUT variants of interest were renal agenesis, renal dysplasia/hypoplasia, hydronephrosis (ureteropelvic junction obstruction or ureterovesical junction obstruction), vesicoureteral reflux (VUR), duplex collecting system, horseshoe kidney, and posterior urethral valves (PUV). Patients without available information of the ultrasonographic parameters were excluded from the analysis. Patients with lithiasis causing hydronephrosis and patients undergoing immunosuppressive therapy were excluded from the analysis.

### 2.3. Urine Sampling

The urine was collected in wide-mouthed sterile containers. Urine was inoculated using a standard calibrated loop, delivering 0.01 mL on chromogenic agar growth medium and Columbia 5% blood agar incubated for 18–24 h at 37 °C. Urinary tract infection (UTI) was defined by positive urine culture with more than 10^5^ colony forming units (CFU)/mL. Bacterial isolation from clinical samples was made using standard microbiological methods. Biochemical characteristics, Gram reactions, and morphology were used to identify the bacterial pathogens from the culture.

### 2.4. Antimicrobial Susceptibility

Urine cultures and antibiotic susceptibility were executed according to standardized protocol and method, using the disc diffusion technique according to the Clinical Laboratory Standards Institute (CLSI) guidelines, on Muller–Hinton agar (MHA), after a standard inoculum adjusted to 0.5 McFarland was swabbed on its surface, using the ready-made antibiotics’ supplied dispenser. All the samples were assessed uniformly in the hospital’s laboratory, which had standard accreditation ISO 15189. Positive urine cultures for uropathogens belonging to Gram-negative Enterobacteriaceae family (GNE group) were tested for thirty-eight antibiotics, and Gram-negative bacteria belonging to Pseudomonadaceae spp., *Stenotrophomonas* spp., and *Ralstonia* spp. (GNP group) were tested for twenty-nine antibiotics, while Gram-positive strains (GP group) were tested for thirty-six antibiotics. Due to laboratory protocol and funding changes in 2016 and 2017, four antibiotics were only tested for urine cultures according to clinical status of patients: ceftibuten, netilmicin, nitrofurantoin, and piperacillin/tazobactam.

### 2.5. Data Collection

The following clinical and demographical data were collected for each patient: age at diagnosis of CAKUT, sex, origin environment, type and side of CAKUT, number of UTIs, type of uropathogen, and uropathogens antibiotic resistance.

### 2.6. Statistical Analysis

Descriptive statistics for numerical variables included means and standard deviations. For categorical variables, frequency as percentage (%) and/or count (*n*) were included. Normal distribution was assessed with the Kolmogorov–Smirnov test. The *t*-test with two factors comparisons was used for variables assuming normal distribution. For variables with non-parametric distribution, the Mann–Whitney test was used. A chi-square test was applied for assessing the demographic features of the patients. All *p*-values of <0.05 were accepted as statistically significant. All data were processed using Statistical Package for the Social Sciences (SPSS) 22 version for Windows (IBM, Armonk, NY, USA).

## 3. Results

During the study period, 252 pediatric patients with CAKUT were admitted to our hospital. We identified 91 that developed a total of 260 episodes of UTI. The remaining 161 patients with CAKUT did not develop UTI events. Out of 91 patients, more than half originated from urban areas (54.91%). Regarding the age at the time of diagnosis of CAKUT, most patients were diagnosed at a young age, within the first year of life (46.15%).

The median age of diagnosing CAKUT was 16 months (range 10 days–17 years and 10 months). Although patient sex distribution was nearly equal (male:female ratio of 1:0.97), more UTI occurred in males (58.07%). This data can be found in Table 1.

### 3.1. Type of CAKUT

We classified CAKUT in three categories: renal malformations (renal agenesis, renal dysplasia/hypoplasia), ureteral malformations (hydronephrosis, ureteropelvic junction obstruction, ureterovesical junction obstruction (UVJO, VUR), and vesical/subvesical malformations (PUV, ano-vesical malformations). Many patients with CAKUT often present with more than one malformation; therefore, a clear separation of malformation type was not possible. Seventeen patients in our study had renal malformation; however, only 9 patients showed solitary renal malformation.

Most patients included in the study had some level of pelvi-ureteral enlargement (80.21%), secondary to PUJO, UVJO, or VUR. The most frequent malformation was unilateral hydronephrosis, identified in more than half of patients (51.64%), while most infections (55%) occurred in patients with bilateral hydronephrosis. They can be found in Table 1. Different grades of VUR were assessed in 26 patients and VUR was responsible for 45.75% of the total number of UTIs (119 episodes of UTI).

### 3.2. Number of UTIs

Out of 91 patients, 84.61% had between 1 and 4 episodes of UTI. Ten patients had between 5–9 infections throughout the study. Only 4 patients developed more than 10 distinct episodes of UTI: two male patients with surgically corrected PUV and VUR with secondary high-grade hydronephrosis and two female patients, one with ano-vesical malformation and bilateral hydronephrosis and one patient with high-grade bilateral hydronephrosis secondary to bilateral VUR (Table 1). Surgical treatment was performed in 41 out of 91 patients in our study. We mentioned the patients with more than 10 episodes of UTI in order to emphasize the problematic subvesical malformations in spite of surgical treatment.

### 3.3. Types of Uropathogens

Two hundred and sixty distinct episodes of UTI were recorded in our study. We classified uropathogens as follows: Gram-negative bacteria (GNE), belonging to the Enterobacteriaceae family represented by: *E. coli, Klebsiella* spp. (*Klebsiella pneumoniae, Klebsiella oxytoca*), *Enterobacter* spp. (*Enterobacter aerogenes, Enterobacter cloacae*), *Proteus* spp. (*Proteus mirabilis*), *Citrobacter* spp. (*Citrobacter amalonaticus*), *Serratia* spp. (*Serratia marcescens*) and *Morganella* spp. (*Morganella morganii*); Gram-negative bacteria (GNP) belonging to the Pseudomonas genus (represented by *Pseudomonas aeruginosa, Pseudomonas fluorescens*), the Stenotrophomonas genus (represented by *Stenotrophomonas maltophilia*), and the Ralstonia genus (represented by *Ralstonia pickettii*); and Gram-positive bacteria (GP): *Staphylococcus aureus, Streptococcus* spp. (*Streptococcus agalactiae*), *Enterococcus* spp. (*Enterococcus faecium*) (Table 2).

The most frequent pathogen causing UTI in children with CAKUT was *E. coli*, followed by *Klebsiella* spp. and *Enterococcus* spp. (Table 2). GNE were the main representatives of causative uropathogens.

UTI occurred slightly more often in male patients (58.07%). The type of germs also seems to have a particular occurrence within the two genders. E. coli was responsible for 55% of female UTIs, while in male patients, E. coli and Klebsiella spp. were likely to have an equal occurrence (27%). Proteus spp. was found more frequently in male patients (11.9% of males, 2.8% of females) (Table 3).

### 3.4. Uropathogens Antibiotic Resistance

GNE group was responsible for 72.69% of all UTIs (189 positive urine cultures). Uropathogens antibiotic resistance was compared over the study period and compared between species within group GNE.

From the penicillin class, amoxicillin/clavulanic acid, ampicillin/sulbactam and ampicillin were found to have statistically significant increased resistance over the years (Figure 1a), (*p* < 0.05).

Overall resistance rates of GNE were relatively high for amoxicillin/clavulanic acid 38.5%, ampicillin/sulbactam 48.1% and 72.3% for ampicillin. Some antibiotics were constantly ineffective in treating the GNE group: cefazoline, cefuroxime, cephalothin, nalidixic acid, and trimethoprim/sulphametoxazole, due to elevated persistent resistance, with a median level over 50%. Amikacin, imipenem, levofloxacin, piperacillin/tazobactam, and vancomycin remain useful antibiotic choices for treating UTI with Enterobacteriaceae, with general sensibility over 70%. Representative substances of all four generations cephalosporins were tested and yearly increasing resistance was questioned. Cefotaxime (*p* = 0.004), ceftazidime (*p* = 0.029), cefepime (*p* < 0.001), and ceftriaxone (*p* < 0.001) developed significant resistance throughout the study, with the highest gained resistance for cefepime and ceftriaxone (*p* < 0.001) (Figure 1b) over the studied period.

Comparing antibiotic resistance of *E. coli* and *Klebsiella* spp., we found overall higher resistance rates for *Klebsiella* spp. (Mann–Whitney, U test, *p* = 0.005), mainly suggestive for gentamicin (*E. coli* 27% resistant, *Klebsiella* spp. 64.8% resistant) and nitrofurantoin (*E. coli* 13% resistance, *Klebsiella* spp. 75.8% resistance), with *p*-values less than 0.001 (Table 4).

GNP was responsible for 24 positive urine cultures causing UTI. Over the years, bacteria in this class slightly enhanced resistance to imipenem (*p* = 0.043), piperacillin/tazobactam (*p* = 0.021), tobramycin (*p* = 0.043), and penicillin (*p* = 0.043). Overall responsiveness was preserved for amikacin (42.5% resistance), quinolones (ciprofloxacin 32.5%, levofloxacin 36.65%), and colistin (0%). Cefepime, ceftazidime, gentamicin, and ticarcillin/clavulanate were constantly resistant to GNP without variable levels on comparative years (*p* > 0.05). Initially, *Stenotrophomonas maltophilia* had sensibility to trimethoprim/sulphametoxazole, but two weeks later, in the same patient, bacteria were completely resistant.

GP group was responsible for forty-seven positive urine cultures that were analyzed. Increased resistance over the years was found for penicillin (*p* = 0.043). Constant high resistant levels were found for clindamycin, erythromycin, gentamicin, quinupristin/dalfopristin, and tetracycline, with frequent maximum levels for several years.

Overall GP group remained susceptible to ciprofloxacin, levofloxacin, ampicillin, and rifampicin in about 50% of cases, while linezolid was suitable for 84.6% of UTI caused by *Enterococcus* spp. Daptomycin, moxifloxacin, and nitrofurantoin remained 100% effective. Classic antistaphylococcal agents vancomycin, teicoplanin, linezolid, and piperacillin/tazobactam persisted effectively in treating urinary staphylococcal infection.

## 4. Discussion

Pediatric patients with obstructed urine outflow tract are at increased risk of developing UTIs, especially when VUR is associated [15,16]. Clinical characteristics of patients with CAKUT include recurrent urinary tract infections, hematuria, and proteinuria in rare cases, but in the majority of cases, patients are asymptomatic [17,18]. Most patients with CAKUT often progress to end stage renal disease (ESRD) at a slower rate; however, certain subgroups do progress more rapidly [19].

The overall prevalence of CAKUT at birth is 3–6 per 1000 live births [19]. CAKUT can occur in patients with other congenital anomalies with a frequency higher than expected in normal population [20]. The prevalence of UTI in patients with CAKUT in our retrospective study was 36%, similar to earlier studies that ranged between 25–59% [21]. Lack of national studies in this area makes our research unique and raises attention on this cluster of patients.

We retrospectively analyzed antibiotic resistance patterns in uropathogens causing UTIs in children with CAKUT over a 5-year period in order to determine levels of antibiotic resistance of uropathogens causing UTI in patients with CAKUT and to establish treatment recommendations based on local patterns.

Although sex–patient distribution was nearly equal, UTIs were more frequent in male patients, similar to other studies in which sex distribution positions male patients at risk of UTI [21]. Urban territory, probably with higher addressability to medical services, accounted for 54.91% of patients, similar to literature data [21].

CAKUT consists of a heterogeneous group of disorders, and the clinical courses of these subgroups are extremely different [19]. Renal anomalies include renal agenesis/hypoplasia/dysplasia or kidney position anomalies such horseshoe kidney or ectopic/pelvic kidney. Often, renal anomalies are not associated with UTIs, and do not increase the risk of UTI, unless associated with other urine outflow abnormalities. However, febrile UTI can cause renal scarring [22]. Outflow abnormalities include both ureteral and vesical malformations [17]. Hydronephrosis was the most frequent defect in our study group (83.51%) caused by PUJO, UVJO, or VUR, the latter being confirmed in 28.57% of cases. In the literature, VUR was associated with an increased risk of non-*E. coli* UTI development [23]. In our study, it could not be established VUR as a risk factor for non-*E. coli* UTI (*t*-test *p* = 0.31), although 45.75% of all UTIs occurred in patients with VUR.

Recurrent UTIs were identified in patients with vesical/subvesical malformation (PUV, ano-rectal malformation) or high-grade VUR and bilateral hydronephrosis in our study. Results are comparable with previous studies that proved lower urinary tract disorder (LUTD) and VUR as risk factors in developing recurrent UTIs [5,16,21,23].

Most patients within the study had between 1 to 4 episodes of UTI. Only four patients had more than 10 episodes of UTI. High-grade bilateral VUR, PUV, or urogenital sinus, in absence of effective surgical treatment, remain challenging cases in terms of antibiotic treatment options [24]. Multiple hospitalizations for UTIs led to increased antibiotic resistance of uropathogens and multidrug-resistant opportunistic strain selection (*Stenotrophomonas maltophilia*, *Proteus* spp., *Serratia marcescens*, *Enterobacter* spp.).

Children with CAKUT seem to have similar causative organisms for UTI with non-CAKUT children; however, there is a higher occurrence of opportunistic bacteria [25]. *E. coli* is responsible for a relatively low number of UTIs in patients with CAKUT in our study (38.84%), compared to the general population, where it stands for 75–90% of childhood UTIs worldwide [5,26]. *Klebsiella* spp. was identified in 21.15% of children with UTI and CAKUT, compared to general pediatric population, where it stands for 10–15% [1,26]. Other UTI causative organisms, such as *Pseudomonas aeruginosa* (8.07%), *Proteus mirabilis* (8.07%), as well as Gram-positive strains (*Enterococcus* spp. 15.76%) have been found with higher prevalence rates within our group compared to general population [26]. *E. coli* was more frequent in female patients (55%), while in male patients, *E. coli* and *Klebsiella* spp. had similar occurrence rates. The bacterial variety between genders has been explained by anatomical differences and related to natural bacterial colonization of the preputial area in non-circumcised boys with non-*E. coli* Gram-negative bacteria [27].

### 4.1. Uropathogens Antibiotic Resistance

A comparison of current guidelines has suggested an optimal approach in treating UTI is to start with wide-spectrum antibiotics (such as amoxicillin/clavulanic acid, third-generation cephalosporins or gentamicin), preferably via oral administration in children older than 2 months of age and adapting treatment 48–72 h later, based on urine culture results [28,29]. Children with high fever, bad general appearance, or vomiting should start treatment intravenously directly [9]. Children with abnormal urinary tract and UTI stand for complicated-UTI and take recommendations of intravenous treatment with wide spectrum antibiotics (amoxicillin/clavulanic acid, cefotaxime/ceftriaxone) [12]. Amoxicillin might be used to cover *Enterococcus* spp. (about 3% of UTI in children) [5]. Where patients are treated with oral antibiotics, empiric selection depends on regional resistance patterns [9,12,30]. In our study, almost half (45.5%) of *E. coli* strains were resistant to amoxicillin, 36% resistant for ceftriaxone and cefotaxime, and 22.8% for gentamicin, the latter remaining valid treatment options, while amoxicillin is debatable. The prevalence of antibiotic resistance in uropathogens, especially *E. coli*, differs significantly between countries, with higher resistance rates reported in some areas [3,31]. Pediatric urinary tract isolates are becoming increasingly resistant to commonly used antibiotics [25].

Nationally implemented reduction of antibiotic misuse, implemented in Northern European countries, lead to lower rates of antibiotic resistance in *E. coli* for amoxicillin/clavulanic acid (10–38%), ampicillin (38–50%), trimethoprim/sulphametoxazole (5–15%) [1,32]. In contrast, countries where antibiotics can be bought over the counter report a higher usage of antibiotics and therefore higher resistance rates: amoxicillin/clavulanic acid (38–40% resistance), ampicillin (78–82% resistance), and trimethoprim/sulphametoxazole (70–85% resistance) [1,27]. Higher resistance patterns were found in London for ampicillin, established to be resistant in 59.2% of cases for *E. coli* and 100% for *Klebsiella* spp. [25]. The percentage of isolates of *E. coli* strains that express resistance to third-generation cephalosporins varies in Europe between 3% (Sweden, Norway) and more than 30% (Slovakia, Cyprus) [33,34]. This increased resistance to primary drugs may be explained due to previous use of antibiotics in general practice and the use of UTI antibiotic prophylaxis in patients with CAKUT [30,35].

In the present study, *E. coli* resistance to third generation cephalosporins varied between 40–60%. In addition, *E. coli* was resistant to amoxicillin/clavulanic acid in 38.5% of cases, to ampicillin in 72.3%, gentamicin 27%, and trimethoprim/sulphametoxazole in 56% of the cases. Amikacin and gentamicin remain valid treatment options for UTIs caused by the GNE group with a relatively low overall resistance of 20–30%. Similar results were observed for quinolones 30–37.5% and carbapenems (imipenem 11.1%).

Uropathogens within the GNP group were responsible for 9.23% of UTI cases in our study, and data were similar to former studies, wherein 11.5% of children in London with underlying renal problems had UTI caused by *Pseudomonas* spp. [25]. In the general pediatric population, occurrence of *Pseudomonas* spp. varies between 1.5 to 2% of UTIs [36]. In our study, the GNP group had high antibiotic resistance for cefepime, ceftazidime, gentamicin, and ticarcillin/clavulanate. Other antibiotics such as amikacin, ciprofloxacin, levofloxacin, and colistin have good results and remain valid treatment options for UTIs with *Pseudomonas* spp. *Stenotrophomonas maltophilia*, usually having intrinsic resistance to antibiotics [37], was in our study initially susceptible to trimethoprim/sulphametoxazole, but in a very short period, it became completely drug resistant.

Gram-positive bacteria were identified in 18.07% of cases of UTIs in our study, and less compared to the London study (29.6%) [25]. In the pediatric population, Gram-positive strains lead to 10.9% of UTI cases. Antibiotic resistance of the GP group was found constantly high for clindamycin, erythromycin, gentamicin, and tetracycline, with levels up to 100% for several years. Other studies support the increased antibiotic resistance for erythromycin (55–71%), clarithromycin (53%), gentamicin (98.5%), ampicillin (31.5–61%), tetracycline (56%), ciprofloxacin (99.2%), and cefuroxime (98.4%) [25,36,38].

Overall increased antibiotic resistance was identified for commonly used antibiotics (ampicillin, amoxicillin/clavulanic acid, and second-generation cephalosporins) between 2016–2020, in the western part of Romania. Lower yet alarming antibiotic resistance was observed for wide spectrum antibiotics (imipenem, meropenem, and vancomycin). The “ESKAPE” group of bacteria remains a challenge for the pediatrician in order to select effective treatment [7]. Antibiotics used for UTI prophylaxis in CAKUT children showed a higher resistance due to extended exposure.

A small number of studies analyzed antibiotic resistance of uropathogens causing UTI in pediatric patients with CAKUT, and revealed a higher percentage of resistance to commonly used antimicrobial agents [29,30]. Therefore, this study, albeit small, offered information regarding antibiotic options for children with CAKUT and UTI based on regional data.

High-grade VUR is by far the most important risk factor for renal scarring [39] and has surgical indication. Patients with CAKUT are referred to the surgeon in our center, similar to the Swiss guideline [24].

### 4.2. Limitations

There are some limitations to this study. Data regarding positive urine cultures performed outside the hospital could not be evaluated. The number of patients in some CAKUT subgroups was small and the subgroups were heterogeneous. Unequal follow-up of patients was caused by the retrospective design of the study. Nonetheless, this study analyzes uropathogens causing UTI in pediatric patients with CAKUT, while many references concern uropathogens causing UTI in general pediatric population. We should not minimize the importance of data that show overall high antibiotic resistance. Due to laboratory protocol and funding changes in 2016 and 2017, four antibiotics were only tested for urine cultures according to clinical status of patients: ceftibuten, netilmicin, nitrofurantoin, and piperacillin/tazobactam. As 2020 was affected by the COVID-19 pandemic, the profile of isolates and antibiotic resistance was influenced. Lower number of samples led to a modified antibiotic resistance pattern.

## 5. Conclusions

Identification of uropathogens’ antibiotic resistance patterns is important in order to create treatment guidelines based on local resistance patterns. While regular updates of these recommendations are necessary, a specific analysis is also needed in patients with CAKUT, as they represent a high-risk group for of UTI and antibiotic resistance. This study is the first Romanian analysis to evaluate uropathogens’ antibiotic resistance in pediatric patients with CAKUT in the western part of Romania.

Our results support interventions that encourage healthcare professionals toward promoting prudent use and careful restriction of antimicrobial drug prescription. Effective treatment for recurrent UTI in patients with CAKUT remains a medical milestone.

## Figures and Tables

**Figure 1 children-08-00585-f001:**
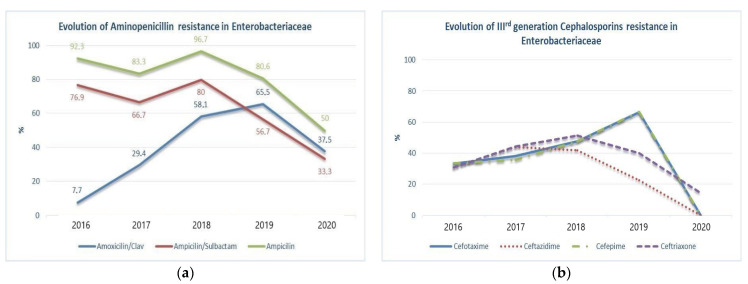
(**a**) Evolution of amoxicillin/clavulanic acid, ampicillin/sulbactam, and ampicillin resistance in Enterobacteriaceae (*p* < 0.001) causing UTI in children with CAKUT; (**b**) evolution of cefepime, cefotaxime, ceftazidime and ceftriaxone resistance in Enterobacteriaceae causing UTI in children with CAKUT.

**Table 1 children-08-00585-t001:** Patient distribution in regard to age, sex, CAKUT type, and UTI frequency.

Number (%)	<12 Months (*n* = 42)	1–3 Years (*n* = 17)	3–6 Years (*n* = 8)	Over 6 Years (*n* = 24)	Total (*n* = 91)
Age distribution	46.15%	18.68%	8.79%	26.37%	100%
Male	29 (69.04%)	9 (52.94%)	3 (37.5%)	5 (20.83%)	46 (50.54%)
**CAKUT Type**					
Renal malformation	4 (9.52%)	1 (5.88%)	1 (12.5%)	3 (12.5%)	9 (9.89%)
Urethral malformation	32 (76.19%)	16 (94.11)	6 (75%)	19 (79.16%)	73 (80.21%)
Vesical/subvesical malformation	6 (14.28%)	-	1 (12.5%)	2 (8.33%)	9 (9.89%)
Number of UTIs					
1 to 4 UTIs	36 (85.71)	14 (82.35%)	7 (87.5%)	20 (83.33%)	77 (84.61%)
5 to 9 UTIs	4 (9.52%)	3 (17.64%)	1 (12.5%)	2 (8.3%)	10 (10.98%)
Over 10 UTIs	2 (4.76%)	-	-	2 (8.3%)	4 (4.39%)

UTI, urinary tract infection; CAKUT, congenital anomalies of the kidney and urinary tract.

**Table 2 children-08-00585-t002:** Annual frequency of uropathogens causing UTI in children with CAKUT, between 2016–2020.

Type of Uropathogens	Total *N* = 260	Percentage (%)	2016 *n* = 21	2017 *n* = 50	2018 *n* = 81	2019 *n* = 77	2020 *n* = 31
**GNE**	189	72.69	17 (80.95%)	34 (68%)	61 (75.30%)	55 (71.42%)	22 (70.96%)
*E. coli*	101	38.84	13	18	31	31	8
*Klebsiella* spp.	55	21.15	3	10	17	21	4
*Proteus* spp.	21	8.07	-	-	9	2	10
*Enterobacter* spp.	6	2.30	1	1	3	1	-
*Serratia* spp.	3	1.15	-	3	-	-	-
*Citrobacter* spp.	2	0.76	-	2	-	-	-
*Morganella morganii*	1	0.38	-	-	1	-	-
**GNP**	24	9.23	-	5 (10%)	12 (14.81%)	3 (3.89%)	4 (12.90%)
*Pseudomonas* spp.	21	8.07	-	4	10	3	4
*Stenotrophomonas maltophilia*	2	0.76	-	-	2	-	-
*Ralstonia pickettii*	1	0.38	-	1	-	-	-
**GP**	47	18.07	4 (19.04%)	11 (22%)	8 (9.87%)	19 (24.67%)	5 (16.12%)
*Enterococcus* spp.	41	15.76	4	11	7	15	4
*Staphylococcus aureus*	4	1.53	-	-	-	4	-
*Streptococcus agalactiae*	2	0.76	-	-	1	-	1

UTI, urinary tract infection; spp., species.

**Table 3 children-08-00585-t003:** Uropathogens causing UTI in children with CAKUT in regard to gender.

Type of Uropathogens	UTI in Males (*n* = 151)	UTI in Females (*n* = 109)	*p* *
*E. coli*	41 (27.2%)	60 (55.0%)	<0.001
*Klebsiella* spp.	42 (27.8%)	13 (11.9%)	<0.001
*Enterobacter* spp.	3 (2.0%)	3 (2.8%)	0.001
*Proteus* spp.	18 (11.9%)	3 (2.8%)	0.001
*Citrobacter amalonaticus*	2 (1.3%)	0 (0.0%)	-
*Serratia marcescens*	2 (1.3%)	1 (0.9%)	0.570
*Morganella morganii*	1 (0.7%)	0 (0.0%)	-
*Pseudomonas* spp.	8 (5.3%)	13 (11.9%)	<0.001
*Stenotrophomonas maltophilia*	0 (0.0%)	2 (1.8%)	-
*Ralstonia pickettii*	1 (0.7%)	0 (0.0%)	-
*Enterococcus* spp.	28 (18.5%)	13 (11.9%)	<0.001
*Staphylococcus aureus*	4 (2.6%)	0 (0.0%)	-
*Streptococcus agalactiae*	1 (0.7%)	1 (0.9%)	-

* Student’s *t*-test; CAKUT, congenital anomalies of kidney and urinary tract; UTI, urinary tract infection; spp., species.

**Table 4 children-08-00585-t004:** Antibiotic resistance prevalence (%) between *Escherichia coli* and *Klebsiella* spp. causing UTI in children with CAKUT.

Antibiotic	*E. coli* %	*Klebsiella* spp. %	*p* *
Ceftazidime	14.9%	42.6%	0.002
Ceftibuten	0.0%	87.5%	<0.001
Ceftriaxone	25.3%	51.9%	0.003
Gentamycin	27%	64.8%	<0.001
Netilmicin	11.8%	44%	0.005
Nitrofurantoin	13.2%	75.8%	<0.001
Piperacillin/Tazobactam	7.4%	24.0%	0.043
Trimethoprim/Sulfamethoxazole	56.6%	73.5%	0.046

* Chi-square test; spp., species; CAKUT, congenital anomalies of kidney and urinary tract; UTI, urinary tract infection.

## Data Availability

Data available on request. The data presented in this study are available on request from the corresponding author. The data are not publicly available due to privacy and ethical restrictions.

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
