# Peer review of "Antibiotic Resistance Patterns of Uropathogens Causing Urinary Tract Infections in Children with Congenital Anomalies of Kidney and Urinary Tract"

_children, 2021, doi:10.3390/children8070585_

Round 1

Reviewer 1 Report

The Authors present an interesting analyses of uropathogens involved in CAKUT UTIs.

The article is well written, the study well conducted.

I just have two observations:

  • How is it possible that only two of your cohort of CAKUT-affected patients underwent surgical correction? The UTIs were before or after surgical correction of CAKUT? Please specify better in the text when and why the patients are referred to the surgeon in your center.
  • There's a recent published study from Parente et Al. [Parente G, Gargano T, Pavia S, et al. Pyelonephritis in Pediatric Uropathic Patients: Differences from Community-Acquired Ones and Therapeutic Protocol Considerations. A 10-Year Single-Center Retrospective Study. Children (Basel). 2021;8(6):436] that reinforces your ideas and conclusions and stresses the attention on the differences between UTIs in uropathic patients and ones acquired in community; I would cite it.

Anyway, as I said before, the work is interesting and I appreciated it.

Author Response

Kind regards,

Florin Horhat.

Reviewer 2 Report

This paper retrospectively evaluated the prevalence of antimicrobial resistance in a single center in Romania in UTI in children with congenital anomalies of kidney and urinary tract (CAKUT).  

Patients with CAKUT are at high risk for UTI and I agree with the importance of the data of antimicrobial resistance of uropathogens of these patients.

However, there are some concerns in this paper as below:

  1. The authors defined UTI as positive urine culture with more than 10^5 CFU/ml in Method section and did not mention whether the patients had some symptoms associated with UTI or not. That seemed just bacteriuria rather than UTI. Some patients with CAKUT frequently show bacteriuria in urine culture, but asymptomatic bacteriuria is less clinically important and not usually treated. How many patients in this study showed symptomatic UTI?
  2. The authors described that they analyzed all patients with CAKUT admitted in their institute over a 5-yearm aged <18 that had at least one episode of UTI and 91 patients met their criteria in Method section. Did they have the patients with CAKUT but without UTI events for the 5 years or not? If have, how many?
  3. The authors described that most of patients were diagnosed as CAKUT within the first year of live, but at the same time, that median age of diagnosing CAKUT was 49.87 months in Result section. In contrast, in Table 1, age distribution of < 12 months and 1-3 years were 46.15% and 18.68% respectively and more than half patients seemed diagnosed CAKUT within 3 years-old. These descriptions confuse the reader.
  4. The authors described out of 91 patients, 84.61% (=77 patients) had between 1 and 4 episodes of UTI, 10 had between 5-9 infections and only 4 patients developed more than 10 episodes of UTI in 3.2. Number of UTI in Result section. In contrast, in Table1, the numbers of patients with UTI events between 1-4, 5-9 and >10 were 62, 11 and 5, respectively. Which are correct?
  5. This paper had only a figure, but in the figure legend the number of the figure was not Figure 1 but Figure 2. Please amend.
  6. Did the figure represent the prevalence of resistant rate in each year? If so, the resistant rate of Ampicilin, Ampicilin/Sulbactam, Cefepim, Ceftazidime and Ceftacime seemed reducing over the years, in spite of the authors’ description. Dunnett’s test seemed more preferable for statistical analysis.
  7. Several numbers or percentages in Table 4 seemed incorrect. (e.g. Netilmicin or Nitrofurantoin resistant E. coli 4 (11.8%) or 7 (13.2%), despite of total number of isolated E.coli was 101. Ceftibuten or Nitrofurantoin resistant Klebsiella 7 (87.5%) or 25 (75.8%), despite of total number of isolated Klebsiella was 55.)

Author Response

Kind regards,

Florin Horhat.

Round 2

Reviewer 2 Report

This revised manuscript was amended according to the reviewer’s comments and the reviewer accepted the authors’ reply from comment 1-5.

In contrast, in response to comments 6 and 7, the author said “some antibiotics were no longer tested for the same pathogen according to the laboratory protocols adapted with clinical status of patient.” And “the laboratory protocols differed over the years. Therefore, different antibiotics were tested for the same uropathogen in different years.” These facts were not described in antimicrobial susceptibility protocol in Methods section. If the protocol were changed in study periods, the reliability of the data of resistant rate over the years can be decreased, but the author should declare the fact and describe criteria of changing the protocol in Method section.

Author Response

Thank you kindly,

Florin Horhat
